# Widespread adaptive evolution during repeated evolutionary radiations in New World lupins

Bruno Nevado[1], Guy W. Atchison[2], Colin E. Hughes[2] & Dmitry A. Filatov[1]

The evolutionary processes that drive rapid species diversification are poorly understood. In particular, it is unclear whether Darwinian adaptation or non-adaptive processes are the primary drivers of explosive species diversifications. Here we show that repeated rapid radiations within New World lupins (*Lupinus*, Leguminosae) were underpinned by a major increase in the frequency of adaptation acting on coding and regulatory changes genome-wide. This contrasts with far less frequent adaptation in genomes of slowly diversifying lupins and all other plant genera analysed. Furthermore, widespread shifts in optimal gene expression coincided with shifts to high rates of diversification and evolution of perenniality, a putative key adaptation trait thought to have triggered the evolutionary radiations in New World lupins. Our results reconcile long-standing debate about the relative importance of protein-coding and regulatory evolution, and represent the first unambiguous evidence for the rapid onset of lineage- and genome-wide accelerated Darwinian evolution during rapid species diversification.

[1] Department of Plant Sciences, University of Oxford, Oxford OX1 3RB, UK. [2] Institute of Systematic Botany, University of Zurich, Zurich 8008, Switzerland. Correspondence and requests for materials should be addressed to B.N. (email: bruno.nevado@plants.ox.ac.uk) or to D.A.F. (email: dmitry.filatov@plants.ox.ac.uk).

Rapid radiations of species are fundamental and ubiquitous components of evolutionary diversification across the Tree of Life in diverse organismal groups and geographical and ecological settings[1–3], and gave rise to many major groups of organisms such as flowering plants[1]. Recent studies are revealing widespread and repeated disparities in rates of species and trait diversification among lineages and through time, documenting an ever-growing number of rapidly evolving and often exceptionally species-rich clades, and shedding light on the extrinsic circumstances and intrinsic traits potentially triggering radiations[4–7]. However, the mechanisms and processes, especially the genomic basis underlying why and how rapid species diversification happens, remain poorly understood.

Rapid diversification is thought to result from both adaptive (for example, ecological speciation) and non-adaptive (for example, geographic isolation) processes[2,8]. When adaptive processes dominate, these lineages are called 'adaptive radiations', while the term 'non-adaptive radiation' has been put forth for rapidly diversifying lineages dominated by non-adaptive processes[8–11]. While most radiations probably involve both adaptive and non-adaptive processes, very few studies have investigated the relative frequency of neutral and adaptive processes in rapidly versus slowly radiating groups[12–14], and the extent to which adaptive changes have wide-ranging effects throughout the genome remains unclear[15]. Instead, most work has focused on rates of species diversification and morphological evolution, and their possible correlations with traits, geological events and ecological shifts[7,16]. However, rates of diversification tell us little about the underlying evolutionary processes and the role of adaptation[8,11], while the rate of morphological evolution is not informative about natural selection acting on non-morphological (for example, physiological or behavioural) traits, which may underpin adaptive radiation but remain undetected[17]. Genome-wide analysis of natural selection at the level of gene sequences and their expression levels, as pursued here, represents a much more direct approach to quantify the role of adaptation in rapid species diversifications.

High rates of DNA evolution, accelerated coding-sequence evolution, accumulation of gene duplications during speciation and sharing of genetic variation between species through ancestral polymorphisms or natural hybridization have been suggested as key genomic features underpinning evolutionary radiations[12,14,18,19]. However, the scarcity of well-sampled comparative studies of genome-wide selective pressures in rapidly and slowly radiating groups of species means that how rapid diversification unfolds at the genetic level remains unclear. In particular, it is unclear whether rapid speciation and ecological and morphological evolution are driven by changes in many genes throughout the genome, or in just a few key regulatory genes. Furthermore, whether changes in protein-coding sequences or expression levels of genes play the predominant role in adaptation is actively debated and remains unresolved[20,21].

The genus *Lupinus* (Leguminosae) provides an ideal model system for understanding the genetic basis of rapid species diversification. It is a species-rich clade of ~280 species[22,23], with a hypothesized ancestral area in the Old World, and two dispersals to the New World within the last 10–12 Myr (refs 23,24). The majority of species diversity occurs in the Americas, with only 13 species in Europe and North Africa. Within the New World, *Lupinus* spp. exhibit very diverse growth forms (ephemeral annuals, prostrate perennial herbs, acaulescent rosettes, stem rosettes, woody shrubs and small trees) and thrive in a very wide array of ecosystems and climates (coastal dunes, chaparral, sagebrush steppe and grasslands, but especially open mountain forests, meadows and disturbed slopes, extending to sub-alpine/alpine elevations) across an exceptionally wide altitudinal range (from sea level to 4,900 m, close to the upper global elevation limit for plant growth), and spanning ~100° of latitude from ~30°S in the Andes to ~70°N in Alaska.

On the basis of extensive taxon sampling and phylogenetic analyses, a set of parallel and nested radiations with significantly higher net diversification rates compared with background levels in the genus (net diversification rate $r = 0.10$–$0.48$ lineages per million years) were identified within the New World: the western North American perennial lupins with 58 species, an estimated age of 4.6 Ma and $r = 0.46$–$1.76$; the eastern South American clade (35 spp, 6.5 Ma and $r = 0.36$–$1.33$), the Mexican and Central American species (46 spp, 2.7 Ma and $r = 1.19$–$6.15$) and the Andean clade (~81 spp, 2.7 Ma and $r = 1.56$–$5.21$)[22–24]. Net diversification rates in these clades are amongst the highest documented for plants[23], exhibit an up to 12-fold increase over background rates observed in *Lupinus*[22], and are apparently still accelerating, suggesting that these radiations may be in the early explosive phase of radiation[25]. The rapid radiations within New World lupins are strongly associated with the derived evolution of perennial life history and dispersal into montane ecosystems[22,24]. It is thought that the shift to perennial habit allowed colonization of higher altitude regions, and enabled the accelerated diversification of growth forms via ecological release in response to ecological opportunities in physiographically heterogeneous montane environments[23,25]. The Andean lupins are particularly notable for being among the most rapidly diversifying plant clades documented to date, with an exceptionally high rate of net species diversification[22], in line with other high-altitude Andean grassland plant radiations which are predominantly young and fast[26], and comparable to rates of diversification documented for cichlid fish radiations in African lakes[23,27]. In contrast to these rapid evolutionary radiations, the annual, lowland species of *Lupinus* exhibit significantly lower species diversification rates, including the Old World species, the eastern North American clade and the western North American annual species[22,24,28].

Here we investigate the role of natural selection during rapid evolutionary radiations in New World lupins. Taking advantage of the striking among-lineage variation in species diversification rates within this genus, we show that rapid species diversification is underpinned by rapid onset of lineage- and genome-wide accelerated adaptive evolution affecting both coding sequences and expression levels of genes. Our results support a preponderant role for adaptive evolution in evolutionary radiations, and reconcile a long-standing debate on the relative contribution of coding and regulatory changes in evolution.

## Results

**Transcriptome sequencing.** To analyse selective pressures at a significant fraction of genes across the genome we obtained transcriptome sequence (RNA-seq) data from 55 New World lupin species (67 individuals totalling 2.6 billion Illumina HiSeq reads, Supplementary Data 1). We included multiple representative species from *Lupinus* lineages with contrasting diversification rates: the slowly diversifying North American annual lineages (NAA, 11 species); and the rapidly diversifying North American Perennial lineages (NAP, 11 species) and the Andean clade (26 species, including nine as yet undescribed species). We also included species representing the other New World lineages: the eastern South American, eastern North American and Mexican lupins. The notable advantage of the RNA-seq approach is the ability to track adaptation at both sequence and gene expression levels, to address long-standing debate about the relative importance of structural and regulatory evolution in adaptation[20,21]. We identified and analysed coding sequences and normalized expression values for 6,013 orthologous genes present

in at least 30 species (Supplementary Fig. 1) including both the rapidly and slowly radiating lineages of lupins.

**Phylogeny and diversification rates.** Phylogenetic reconstruction using maximum-likelihood (ML) on the concatenated data set and coalescent-based species tree methods returned very similar results (Supplementary Fig. 2 and Supplementary Note 1). Henceforth, we discuss only the results obtained with the coalescent-based approach, but results obtained with the ML phylogeny were similar throughout and are reported in Supplementary Files.

The phylogeny obtained (Fig. 1) is largely congruent with previous studies[22,23], but exhibits robustly supported species

relationships even within the very young Andean clade that previously lacked resolution (Supplementary Fig. 2). Phylogenetic reconstruction using only gene expression values resulted in a very similar topology (Supplementary Fig. 3). Estimated diversification rates confirm the previously reported increase in diversification rate in NAP, Mexican and Andean lineages[22,23] (Supplementary Fig. 4).

**Selection on coding sequences.** To estimate the frequency of natural selection on coding-sequence evolution, we implemented dN/dS phylogenetic tests for each gene. As the power of these tests depends on the number of species used in the analysis[29], we

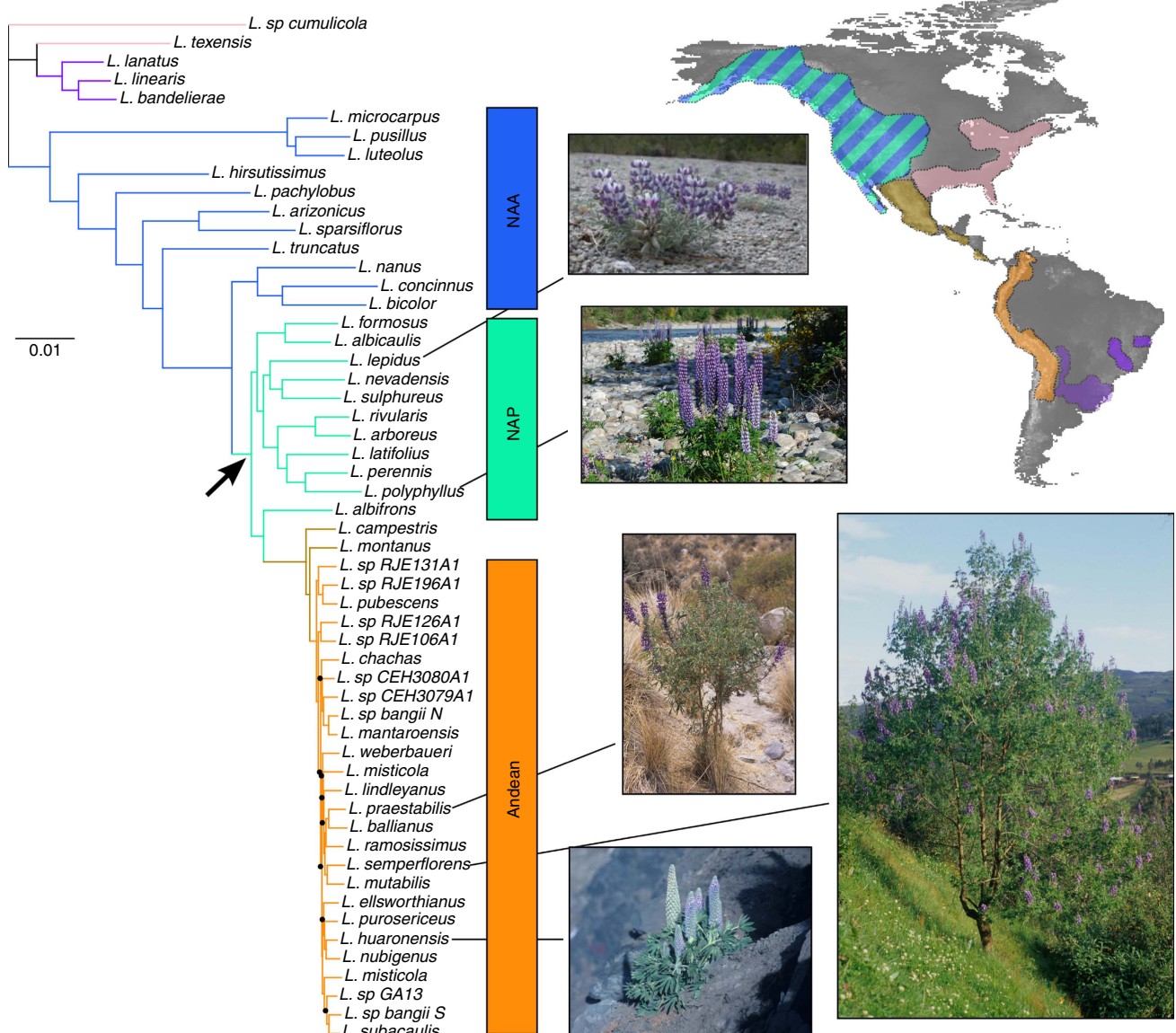

**Figure 1 | Coalescent-based phylogeny of New World lupins based on 6,013 loci.** The black arrow marks the shift from annual lowland species to mostly perennial montane lupins, which coincided with a significant increase in net species diversification rate. All nodes have >90% bootstrap support except those marked with black dots. Inset photographs illustrate the diversity of growth forms across the genus, which includes ephemeral annuals, prostrate perennial herbs, acaulescent rosettes, stem rosettes, woody shrubs and small trees (photos by C.E.H.). Inset map shows the geographic distribution of New World lupins, encompassing a variety of ecosystems and climates (coastal dunes, chaparral, sagebrush steppe and grasslands, but especially open mountain forests, meadows and disturbed slopes, extending to sub-alpine/alpine elevations) across an exceptionally wide altitudinal range (from sea level to 4,900 m). Inferred distribution ranges of the main *Lupinus* lineages are coloured to match branches on the phylogeny (adapted from ref. 22). The main subgroups studied are highlighted: the NAA, NAP and the Andean clade. Tree scale is in expected substitutions per site.

analysed separate subgroups of 10 randomly selected species of slow (NAA) or rapid (NAP and Andean) lineages, and the entire New World and Andean clades.

The percentage of genes evolving under positive selection ranged from 5 to 32% in different *Lupinus* lineages (Fig. 2a and Supplementary Fig. 5). We found a clear difference between slowly and rapidly diversifying lineages, with only 5.8% of positively selected genes in NAA, while 13.1–16.8% showed evidence of positive selection in the rapidly diversifying lineages (Supplementary Table 1). This result is robust regardless of the sub-sample of Andean species analysed (ANDs1 and ANDs2, Fig. 2a,b). For genes showing evidence of positive selection, the

percentage of sites affected ($P_{sel}$) and the dN/dS estimates for these sites ($\omega_1$) varied widely within and between groups analysed, and were very high in some cases ($P_{sel} > 20\%$ or $\omega_{1>} 100$; Supplementary Fig. 6). Estimates of these parameters for data sets with low species divergence, as in our case, can be relatively imprecise, hence we focus on the likelihood ratio tests (LRT) that are robust even if accurate parameter estimates are not possible[29]. Our results are consistent with the higher power for larger sample size (Fig. 2b), and are conservative given that the older age of the NAA subgroup should increase the power[29], yet we see least positive selection in NAA. Similar results were obtained with the branch-site modelling framework (Supplementary Fig. 7 and Supplementary Note 2): the rapidly diversifying Andean clade showed higher rates of positive selection compared with NAA, while NAP exhibited inter-mediate values. Taken together, our analyses reveal a strong signal of accelerated protein-coding evolution associated with rapidly diversifying lineages, with up to three times as many genes evolving under positive selection compared with slowly diversifying lupins. Gene Ontology (GO) analysis revealed no over-represented terms in the set of genes under selection within each group, suggesting natural selection affected many different genes and gene functions.

The percentage of genes under selection, 40% when all 26 Andean species are considered, is strikingly high (Fig. 2b), particularly given the young age of this clade. Although no data sets of comparable size exist for other plant or animal genera to set this result in a wider context, population genetic studies indicate that over 50% of non-synonymous mutations are fixed by positive selection in most species analysed[30–32]. Conversely, the percentage of non-synonymous mutations fixed by positive selection is close to zero in humans[33] and several plant species studied[34,35]. However, both sample size and methodology of analysis vary widely across studies, making direct comparisons difficult. To address this we collected and analysed previously published transcriptome data for seven other plant genera, each represented by six to eight species (Supplementary Data 2). To account for sample size differences we re-analysed random subsets of eight species from rapidly and slowly diversifying *Lupinus* lineages. We find that the percentage of genes evolving under positive selection in the slowly diversifying NAA lupins is similar to other plant genera, while the high frequency of positive selection within rapidly diversifying NAP and Andean lupins remains unmatched (Fig. 2c and Supplementary Table 2).

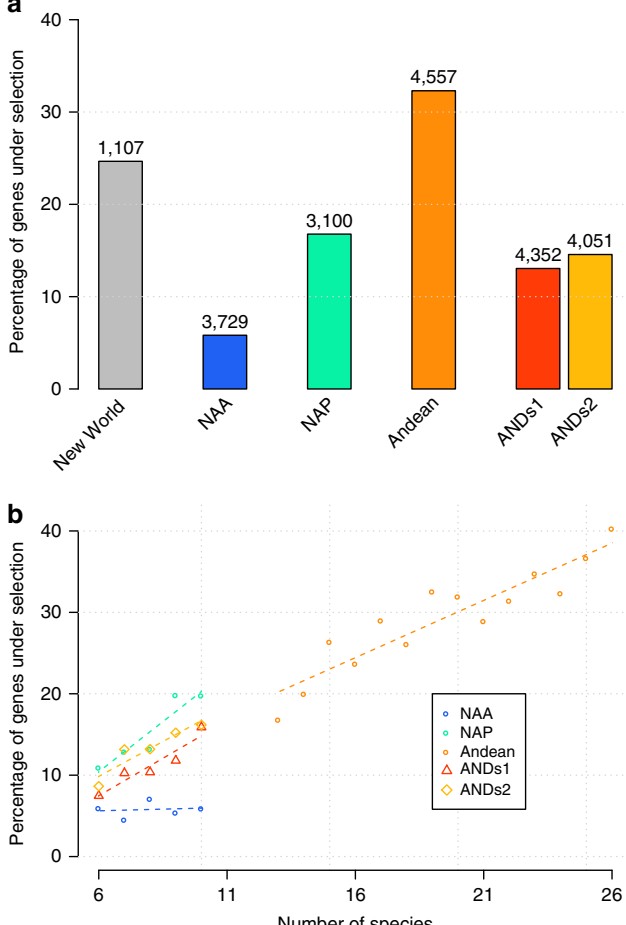

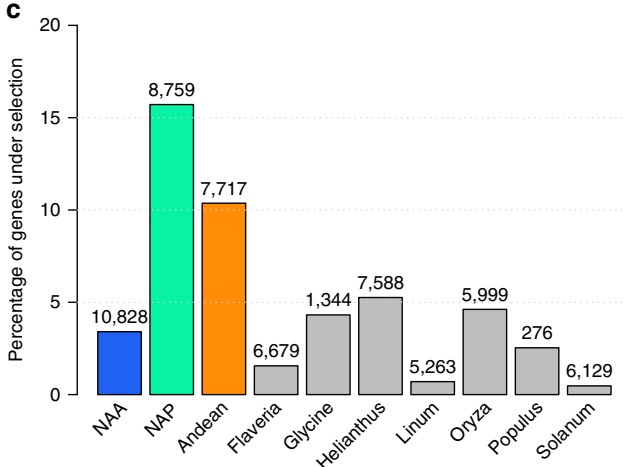

**Selection on gene expression levels**. To understand the evolution of gene expression levels during rapid diversification, we calculated a rescaled relative branch length index for each branch (RRBL; Fig. 3), which reflects overall changes in gene expression

**Figure 2 | Phylogenetic analysis of selection on coding sequences.** (**a**) Percentage of genes evolving under positive selection in the different *Lupinus* lineages (New World—all species; NAA; NAP; ANDs1 and ANDs2 are random subsets of 10 Andean species). (**b**) The effect of the number of species used on the power to detect selection. Because of missing data, different genes were sampled in a variable number of species. The graph shows the percentage of genes inferred to contain sites evolving under positive selection, for each group and according to the number of species sampled. Results are only shown for points where 100 or more genes were tested. (**c**) Percentage of genes evolving under positive selection across seven plant genera with transcriptomes available for sets of species, compared with the different *Lupinus* lineages studied. Each group was tested using between six and eight species. For **a** and **c** values depicted are averages across all genes sampled in at least six species, and numbers on top of bars denote number of genes tested for selection with each group.

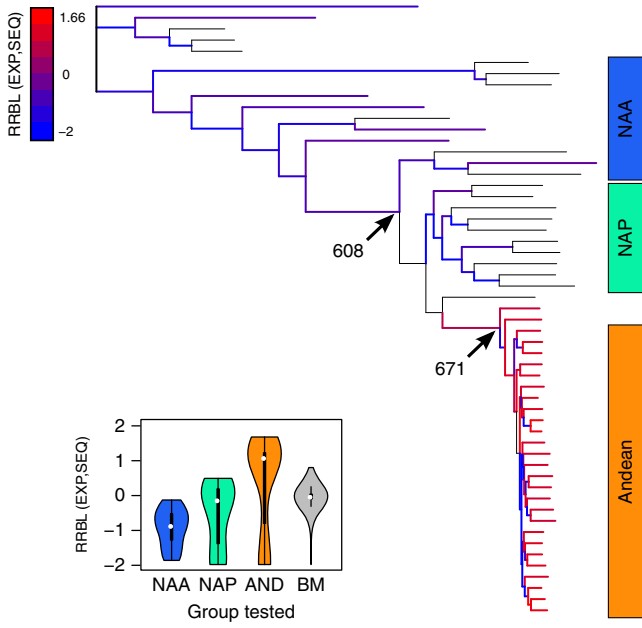

**Figure 3 | Evolution of gene expression in New World lupins.** Branches are coloured according to the relative contribution of gene expression and sequence divergence (measured with RRBL): high values indicate increased divergence in gene expression values. Branches with RRBL values within the 2.5th and 97.5th percentiles of the simulated distribution under BM (no selection on gene expression) are coloured black. Inset graph shows the distribution of RRBL values per branch in the different lineages (NAA; NAP; and Andean clade) and for data simulated under a Brownian motion model (that is, without selection on gene expression values, denoted BM). Black arrows indicate the two nodes with the highest number of shifts in optimal expression values of individual genes (number of genes preferring shift at each node are indicated), based on analysis of selection using Ornstein–Uhlenbeck models.

values relative to sequence-based divergence along each branch. We obtained the expected distribution of RRBL values under a strictly neutral model of gene expression evolution by simulating gene expression changes along the observed phylogeny with Brownian motion (BM) processes. We found that the Andean clade is significantly enriched for positive RRBL values (two-sided binomial test, $P = 2.294 \times 10^{-6}$), and mean RRBL values were highest in the Andean lineage, lowest in the NAA and intermediate in the NAP (inset Fig. 3 and Supplementary Fig. 8). As we used the observed tree topology and branch lengths in the simulations and did not detect any clustering of high RRBL values, our results are unlikely to represent an artefact of the shorter branch lengths in the Andean clade, but instead indicate more frequent directional changes in gene expression values in this clade, possibly driven by positive selection. Results of RRBL analyses using subsets of species for which the same plant tissues (see Supplementary Data 1) were collected, returned identical results (Supplementary Fig. 8). Thus, the observation of higher expression divergence in the Andean clade is also robust to the slight differences in tissue sampling between species.

To detect whether shifts in gene expression values occurred gradually and randomly, or in punctuated bursts during diversification of New World lupins, we performed Ornstein–Uhlenbeck (OU) tests on gene-expression values. These tests revealed that a model including at least one shift in optimal expression level was preferred for most genes (5,108; 93% of genes tested), and that these shifts were strongly clustered: 671 (13%) occurred close to the base of the Andean clade, subtending the fastest diversifying lupin lineage[22,23]; while a further 608

(12%) occurred near the branch leading to the NAP (Fig. 3), closely matching the shift to mostly perennial life cycle and montane habitats. These two nodes appear as outliers in the distribution of gene expression shifts along the phylogeny (Supplementary Fig. 9). Similar to the analysis of RRBL values, clustering of gene expression shifts in these nodes is robust to differences in tissue sampling between species (Supplementary Fig. 9). Gene Ontology analysis revealed several over-represented terms in genes showing shifts in gene expression in outlier nodes, including cellular components (GO0009579 thylakoid, GO0009536 plastid), molecular functions (GO0003824 catalytic activity, GO0004518 nuclease activity and GO0016788 hydrolase activity) and biological processes (GO0015979 photosynthesis) (Supplementary Fig. 9).

## Discussion

Our results represent the first systematic genome-wide analysis of natural selection at the molecular level across closely related congeneric lineages with sharply contrasting slow and fast species diversification rates. With dense sampling of species across lineages, we demonstrate that explosive species diversification in lupins is underpinned by widespread adaptive evolution at the protein sequence and gene-expression levels, indicating the importance of both processes for rapid diversification. Accelerated protein evolution is not an inherent feature across all lupins, as the slowly diversifying annual species exhibit levels of selection comparable to other plant genera. Instead, the increase in frequency of positive selection is restricted to the rapidly diversifying *Lupinus* lineages, illustrating the involvement of widespread Darwinian selection during rapid species diversification.

Accelerated coding-sequence evolution during rapid diversification has been suggested in other evolutionary radiations of both animals and plants. In cichlids, comparison of four East African species to the Nile tilapia (*Oreochromis niloticus*) showed higher dN/dS values for species belonging to the rapidly diversifying clades, but failed to distinguish a scenario of relaxed purifying selection from increased positive selection[12]. Furthermore, this study included only one or two species from each of the East African Lakes (Tanganyika, Malawi and Victoria), and a single closely related representative of the 'slow radiating' *O. niloticus* lineage—a lineage encompassing over 60 species and including several rapidly diversifying sub-lineages, and thus not necessarily the ideal representative of a slowly diversifying clade[17]. Similarly, in birds higher than average dN/dS values and an increase in the number of genes under positive selection were reported in the silvereye, *Zosterops lateralis*[14], but this analysis used a single representative species of the large and rapidly diversifying whiteye bird radiation, and compared it to distantly related species (over 20 Ma divergence). In plants, a high percentage of genes under selection was detected for species in the Hawaiian *Schiedea* radiation compared with other plant genera[13], but the number of genes analysed (36) was small. Furthermore, a transcriptome-wide analysis of the wild tomato clade detected several genes experiencing positive selection[18], but the authors did not compare these values to related lineages with lower diversification rates and thus the link between positive selection and diversification rates remained unclear. Our results thus represent the first unambiguous example of a rapid shift between low and high rates of coding-sequence evolution concomitant with a large increase in species diversification rates, and affecting a large number of species within the same lineages. Similar analyses of densely sampled phylogenies of other evolutionary radiations will be necessary, to conclude whether accelerated adaptive evolution is a general property of rapid species diversification.

The genome-wide effect of Darwinian selection during rapid diversification is illustrated by our finding that over 40% of genes have undergone adaptive evolution of coding sequences during the recent radiation of Andean lupins. The Andean clade spans almost 5,000 m of elevation with a few species at low altitudes and cohorts of different species at mid-, high- and very high altitudes, including a suite of species adapted to the extreme daily freeze-thaw conditions found between 4,500 and 4,900 m, close to the upper limits for plant growth. The high percentage of genes under selection in this clade suggests that rapid diversification in Andean lupins was at least partly driven by ecological divergence, as species adapted to the very diverse habitats spanning the recently formed and physiographically heterogeneous Andean mountains, in line with recent work showing a large role for ecologically driven diversification in other Andean clades[18].

We found close associations between clusters of shifts in gene expression and shifts in species diversification rates, habitats and life history traits, suggesting that rapid gene expression divergence may underpin rapid evolutionary diversification. Interestingly, the very young Andean radiation shows a higher rate of expression divergence than the older NAP species. Both lineages show increases in net species diversification rates compared with annual lupins (NAA), but phenotypic diversity and the rate of morphological diversification are both highest within the Andes, with spectacular variation in plant habit and size, showing a 100-fold variation in plant height and ranging from minute prostrate mat-forming plants less than 5 cm tall, to acaulescent cushion-like perennial herbs, to large stem rosettes, to woody shrubs and treelets 5–6 m in height and 10 cm stem diameter. Variation in growth forms and plant size is matched by similar >100-fold diversification in inflorescence size, disposition and form. The association between gene expression divergence and rates of morphological diversification supports the idea that gene expression plays a dominant role in morphological evolution, a theory first proposed more than 30 years ago[36] and recently supported by empirical studies in model organisms[21,37,38], but hitherto untested in non-model organisms or evolutionary radiations.

The phylogeny obtained in this study supports the paraphyly of the North American Perennial lupins with respect to the Mexican and Andean lineages. This suggests that North American perennial lupins represent a first foray into montane environments, conferring pre-adaptation to montane conditions in lineages that subsequently dispersed to the Mexican and Andean mountain ranges where they diversified rapidly and extensively[22], a pattern seen in other plant groups pre-adapted to cold environments in mountain ranges such as the Andes[39]. In this regard, the New World lupin radiation is similar to that of the large adaptive radiation of cichlid fish in the East African Great Lakes, where the older Lake Tanganyika harbours a paraphyletic assemblage from which a lineage, the Haplochromini, emerged to colonize and radiate extensively in Lake Malawi and Lake Victoria[40,41]. The extremely high diversification rates in the latter two lakes are thus derived from lineages pre-adapted to lacustrine environments that originally evolved in Lake Tanganyika[42,43]. Together with the steep multi-dimensional ecological gradients found in both lakes and mountains (for example, in depth or elevation, water or air temperature, substrate or soil type), the arrival of species pre-adapted to available habitats seems a crucial factor driving the very fast diversification rates seen in such disparate groups as cichlids and lupins[23,27].

Studies of other evolutionary radiations suggest an important role for genomic mechanisms that provide evolutionary flexibility and allow clades to diversify rapidly when new ecological opportunities arise. In particular, the recruitment of genes from gene duplication events, and the sharing of genetic variation between species through ancestral polymorphisms or natural hybridization, have been proposed as important drivers of rapid species diversification[12,14,18,19]. Chromosome rearrangements and genome duplication may have played a significant role in evolution of lupins, as chromosome numbers vary considerably throughout the genus ($2n = 30$ to $2n = 52$) (ref. 22). However, this source of evolutionary novelty may be less significant in the species studied in this paper because chromosome number is constant ($2n = 48$) across the whole western New World clade encompassing the North American Annuals, North American Perennials, Mexican and Andean species. Nevertheless, the role of gene and genome duplication in adaptation will be an interesting area for future research.

## Methods

**Data collection.** We obtained *Lupinus* spp. samples from different sources listed in Supplementary Data 1. We included multiple representative species from *Lupinus* lineages with contrasting slow or rapid diversification rates. To maximize the sampling of genes, we collected and homogenized different plant tissues; stem and leave tissue were available for all samples, while flower tissue was collected only for species that flowered in the greenhouse (Supplementary Data 1). We sampled young leaves with stem, and flowers from developing buds. For each accession tissues were collected from a single plant, up to 100 mg of tissue was flash-frozen in liquid nitrogen, ground to powder and used for total RNA extraction with the RNAeasy Plant extraction kit (Qiagen), following the manufacturer's instructions and performing the optional DNase digestion step. The integrity of extracted RNA was assessed with the Agilent 2100 Bioanalyzer, and samples with low integrity values were re-extracted. For each sample, paired-end Illumina libraries were constructed, samples multiplexed and sequencing performed on the Illumina Hi-seq sequencing platform with paired-end reads of size 100–125 bp.

**Transcriptome assembly.** We quality checked the short sequence reads of each species using FASTQC v 0.11 (available from http://www.bioinformatics.babraham-m.ac.uk/projects/fastqc). We used CUTADAPT v 1.4 (ref. 44) to remove illumina adaptors, and TRIMMOMATIC v 0.32 (ref. 45) to trim reads. We trimmed leading and trailing bases with quality below 5, and cropped reads if the average quality per base dropped below 15 over 4 consecutive bases. We discarded reads smaller than 36 bp after trimming. To assemble a *de novo* transcriptome for each individual, we used TRINITY v r20140413p1 (ref. 46) with default settings except for the minimum assembled contig length (set to 300 bp). We identified putative coding sequences (CDSs) within each transcript with TRANSDECODER (available from https://transdecoder.github.io/) using default settings.

**Orthology inference.** Using a clustering and phylogeny-based orthology inference method[47], we identified 6,013 orthologous genes longer than 300 bp present in at least 30 samples (Supplementary Methods). Each orthologous gene was annotated with GO terms in Blast2GO (ref. 48) using the NCBI non-redundant database (*E*-value threshold of $10^{-5}$ and considering only hits with minimum 50% coverage). GO enrichment tests for over-represented terms were performed using Fisher's exact tests at false discovery rate (FDR) corrected $P < 0.05$ (ref. 49). Coding sequences of these 6,013 orthologous genes and their relative gene expression levels were used in all subsequent analyses.

**Phylogenetic inference.** To recover the phylogenetic relationships within the New World lupins sampled in this study, we employed three alternative approaches (Supplementary Methods): ML phylogenetic inference[50] on the concatenated data set of all orthologs (supermatrix approach); coalescent-based phylogenetic reconstruction[51] using individual gene-trees (species tree approach); and neighbour-joining based on the gene expression values. For all phylogenetic analyses we used only orthologous genes available in at least 30 species and with 300 bp remaining after trimming positions with more than 30% missing data.

**Phylogenetic conflict.** To infer the extent of phylogenetic conflict among genes, we performed the Shimodaira-Hasegawa test (SH-test)[52] with each gene to compare the fit of an unconstrained gene tree to either of the two species trees (supermatrix or species trees). We used PHYML v 3.1 (ref. 53) with the GTR + Γ model to optimize the likelihood of the unconstrained gene tree and the two species trees. We then used the individual site likelihoods reported by PHYML to perform the SH-test in CONSEL v 1.2 (ref. 54), and excluded genes that significantly preferred the gene tree over the species tree (FDR-corrected $P < 0.05$).

**Diversification rates.** To test whether previously detected net diversification shifts within New World lupins are supported in our data set, we performed a Bayesian analysis of macroevolutionary mixtures[55]. We converted the sequence-based phylogenies obtained previously into ultrametric trees using the

R (https://www.r-project.org/) package APE v 3.2 (ref. 56). We used the function CHRONOPL, which implements the penalized likelihood method[57]. We arbitrarily set the age of the root to 1 (we were only interested in relative diversification rates), and estimated the optimal value for the rate smoothing parameter by cross-validation. We used the resulting ultrametric tree in the program BAMM (ref. 55) to obtain estimates of net diversification rates for each branch with the speciation and extinction model. We used previously published estimates for the number of species within each lineage[22] to set clade-specific sampling probabilities as follows: NAA = 0.41 (11 species sampled, 27 estimated); NAP = 0.17 (11/63); Andean clade = 0.32 (26/81); Mexican clade = 0.04 (2/46); and eastern New World species = 0.1 (4/39). To test for convergence, we ran and compared the results of three independent MCMC simulations, 10 million generations each, sampling every 5,000 generations. The R package BAMMTOOLS v 2.0 (ref. 58) was used to test for convergence, plot the estimated net diversification rates per branch, and identify the most credible set of distinct rate shift configurations.

**Selection on coding sequences.** To estimate the frequency of positive selection acting on coding sequences, we used methods based on the estimation of the non-synonymous to synonymous substitution rate ratio ($\omega =$ dN/dS) along phylogenies[59,60]. We analysed each gene using 'sites models'—which allow $\omega$ to vary among sites within each gene—and 'branch-site models', which allows $\omega$ to vary among sites of the gene and branches of the phylogeny. We performed these analyses using only genes that did not show significant phylogenetic conflict (SH-test).

We used the program CODEML from the package PAML v 4.8 (ref. 61) to fit to each gene the 'sites models' M7 and M8 (ref. 59). Both models split the sites in an alignment into classes: model M7 considers only two classes of sites, viz. neutrally evolving ($\omega_1 = 1$) or negatively selected sites ($\omega_0 < 1$); while model M8 additionally includes a class of sites allowed to evolve under positive selection ($\omega_S > 1$). We compared the fit of the two models with a LRT with 2 degrees of freedom, and corrected resulting $P$ values with the FDR method. For genes significantly rejecting M7 in favour of M8, we performed a second test, comparing the fit of models M8 and M8a. The latter has the same parameters as M8 but $\omega_S$ is fixed to 1 (ref. 62). We classified genes as evolving under positive selection when model M8 exhibited a significantly better fit than model M8a (LRT with 1 degree of freedom, FDR corrected $P < 0.05$). We performed these 'sites-models' tests using the entire data set of New World lupins and the entire Andean assemblage, as well as separately for subgroups of 10 randomly selected species representing the slow radiating North American Annuals (NAA) and the fast radiating North American Perennials (NAP) and Andean lupins (including two random subsets of 10 species denoted ANDs1 and ANDs2). For each test, we removed sequences and codons with more than 80% missing data. Genes with less than six species or 99 codons were excluded, as were genes without variation, and genes with too high synonymous substitution rates that could indicate alignment errors (we used a dS threshold of 2, but different thresholds provided identical results). For all tests, equilibrium codon frequencies were estimated from the average nucleotide frequencies at the three codon positions.

To identify genes containing sites that experienced episodic positive selection on specific lineages we used the adaptive branch-site random effects likelihood method[63] (aBS-REL) implemented in the package HYPHY v 2.2 (ref. 64). This method fits a variable number of site classes—some of which are allowed to experience positive selection—to each branch of the phylogeny. Using a model-selection procedure the method infers the optimal number of classes for each branch, and tests whether any of these classes is evolving under positive selection ($\omega > 1$). We used the same set of filters as for the 'sites-models' described above, and performed these tests for each gene in the data set including all New World lupins. The branch-site tests returned a list of branches under selection in specific genes. Because of different species being sampled on different genes, it was sometimes impossible to unambiguously assign specific gene-branch combinations to individual branches in the full topology. We refer to unambiguously mapped branches as those where both nodes defining the branch were present in the gene and the species tree (Supplementary Fig. 10).

**Selection on gene expression levels.** To understand the evolution of gene expression during the diversification of New World lupins, we performed two complementary analyses using the normalized gene expression levels of 6,013 orthologous loci (Supplementary Methods): one comparing branch lengths based on expression divergence with branch lengths based on DNA-sequence divergence; and a second model-selection approach, which allowed us to pinpoint nodes in the phylogeny exhibiting a large number of shifts in optimal gene expression values. While biological replicates are desirable when studying gene expression differences between species, for most species multiple seed accessions collected in the field were not available due to the difficulties of collecting viable *Lupinus* seed from remote locations (for example, from the Andes). While this introduces additional noise to gene expression estimates, it does not bias our results in any of the analyses performed. Furthermore, due to dense species sampling in our study, gene expression values for closely related species act as biological replicates for internal branches of the phylogeny.

To assess the relative contribution of sequence- and expression-based divergence on each branch, we calculated a RRBL index for each branch. We first

calculated gene expression distances between pairs of samples as $1 - \rho$, where $\rho$ is Spearman's correlation coefficient[65] between normalized gene expression values. Using the gene expression distance matrix, we re-estimated the branch lengths of the sequence-based trees using the Fitch–Margoliash method implemented in the program FITCH v 3.6 (part of the PHYLIP package[66]), while disallowing negative branch lengths. We rescaled the resulting branch lengths in the expression trees such that the sum of branch lengths across both expression and sequence-based trees was the same. For each branch we then calculated a RRBL scalar

$$RRBL[EXP, SEQ] = \frac{BL[EXP] - BL[SEQ]}{mean(BL[EXP], BL[SEQ])}, \qquad (1)$$

where $BL_{[EXP]}$ is the rescaled branch length from expression data and $BL_{[SEQ]}$ is the corresponding branch length in the sequence-based tree.

Positive values of $RRBL_{[EXP,SEQ]}$ indicate branches with accelerated gene expression divergence compared with sequence divergence, while negative values indicate the opposite trend. To test the significance of RRBL values different from 0, we performed a simulation study. We simulated gene expression values for each gene using a BM process across the sequence-based phylogeny, using the R package OUCH v 2.8 (available from http://cran.at.r-project.org/web/packages/ouch/). Parameters of the BM process were set to the empirical values for each gene. We analysed the resulting simulated data set as before to estimate RRBL values for each branch, and summarized RRBL values across branches from 100 simulated data sets. The resulting distribution reflects the expected distribution of RRBL values if the variance in gene expression values observed across species was due solely to random drift, and can be used to detect significant deviations in the observed values (for example, which observed RRBL values fall outside the 95% simulated distribution).

To test whether shifts in optimal gene expression values occurred gradually and randomly during the diversification of New World lupins or were clustered on specific branches of the phylogeny, we used the OU modelling framework[67,68], implemented in the R package OUCH, on the normalized expression levels of ~5,500 orthologous genes (for ~500 genes OU-models failed to converge). OU processes are routinely used to model the evolution of quantitative traits along phylogenetic trees, of which gene expression values are one example. OU processes can accommodate different degrees of drift ($\sigma^2$) and selection strength ($\alpha$), and can be used to infer optimal trait values ($\theta$). In addition, models allowing for a different number of OU processes along phylogenies can be compared using a LRT to infer shifts in optimal trait value in specific branches. For each gene, we compared an OU model allowing for a single optimal expression level for all the branches of the phylogeny (OU1-model) to models that allow a shift in the optimal expression value in a specific node of the phylogeny (OU2-models). We fitted OU2-models to each possible node in the phylogeny, and compared the likelihood of the best-fitting OU2-model to the OU1-model, using a LRT with 1 degree of freedom. For genes significantly rejecting the OU1-model compared with the best OU2-model (FDR corrected $P < 0.05$), we recorded the node on which the shift in optimal gene expression level occurred. We then summarized how many genes changed gene expression levels on each node of the phylogeny. For species with more than one accession available, for each gene we used the mean expression level of all conspecifics. This mean-approach was recently shown to have negligible effect on both power and FDR to detect shifts in gene expression using the OU modelling framework[69]. To further reduce the noise introduced by having a single expression level measurement for most species, we only considered OU2 models that included at least three species in each of the lineages defined.

**Data availability.** RNAseq data that support the findings of this study have been deposited in the NCBI Sequence Read Archive (SRA) with the accession codes SAMN04869551 to SAMN04869616. Other RNAseq data referenced in this study are available in GenBank with the accession codes SAMEA3178181 (*Lupinus polyphyllus*); SRR1141027, SRR1141029, SRR1141037, SRR1141040, SRR1165196, SRR1165206, SRR1166363 and SRR1166375 (*Flaveria* spp.); SRR1171830, SRR1171850, SRR1174226, SRR1211016, SRR452306 and SRR452313 (*Glycine* spp.); SRR1041637, SRR1043153, SRR1685780, SRR826582, SRR826604, SRR826626 and SRR826631 (*Helianthus* spp.); SRR957658-SRR957663, SRR957665 and SRR957669 (*Linum* spp.); SRR1170762, SRR1171631, SRR1174772, SRR1178922, SRR1179195, SRR1220645, SRR1592601 and SRR1685731 (*Oryza* spp.); ERR260313, SRR1046128, SRR1508229, SRR1660793, SRR540223 and SRR901769 (*Populus* spp.); ERR185927, SRR1087902, SRR1104128, SRR122109, SRR1291243, SRR521349, SRR786556 and SRR799447 (*Solanum* spp.).

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

## Acknowledgements

This work was supported by grants from the Natural Environment Research Council (Grant NE/K004352/1 to D.A.F.) and the Swiss National Science Foundation (Grant 31003A_135522 to C.E.H.). We thank the staff at Edinburgh Genomics (University of Edinburgh), the Functional Genomics Centre Zurich (University of Zurich) and the Wellcome Trust Centre (Oxford) for sequencing and computational support, and acknowledge the use of the University of Oxford Advanced Research Computing facility in carrying out this work (http://dx.doi.org/10.5281/zenodo.22558). We thank the United States Department of Agriculture and the Desert Legume Program (University of Arizona) for seedlots, the national authorities in Argentina, Bolivia, Ecuador, Mexico and Peru for permission to collect plant material, Rayko Jonas and Markus Meierhofer (Zurich) and Tim Wroe (Oxford) for help with growing plants. We thank O. Osborne for assistance in the analysis of non-lupin genera, and M. Chester for guidance in the lab.

## Author contributions

D.A.F. conceived the study. B.N., C.E.H. and D.A.F. designed the research and wrote the manuscript. B.N. and G.W.A. generated the data. B.N. analysed the data supervised by D.A.F. All authors discussed the results and commented on the manuscript.

## Additional information

**Competing financial interests:** The authors declare no competing financial interest.

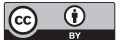

