## [Peer Review File · Nature Communications]

Reviewers' comments:

Reviewer #1 (Remarks to the Author):

Nevado and coauthors present some very interesting data and patterns of shifts in molecular evolution and gene expression in relation to shifts in diversification. In my opinion, however, the descriptions of the data, analyses and conclusion - "our results represent the first unambiguous evidence for lineage-wide accelerated protein-coding evolution concomitant with the onset of rapid evolutionary radiation" - provided in the main text of the submitted manuscript is too vague for most readers to comprehend and evaluate the implications of the findings. The data and analyses are fairly well described in the Supplementary Notes file, but I think there are some important points should be moved from the Supplementary Notes to the main text. Further, it seems to me that the authors could provide the readers with a better understanding of the implications of their findings.

I offer the following more specific comments/suggestions:

1. The authors do not comment on the possible influences of natural hybridization, polyploidy or even gene duplication on their findings. Could any of these processes be contributing to shifts in diversification, molecular evolution or gene expression levels?
2. ~L60 - Readers will want to know what tissues and developmental stages were used for RNA Seq analyses without having to go to the Supplementary Notes. Even after consulting the Supplementary Notes, it is not clear to me that the data are easily comparable across species. I suggest the authors make a simple statement about the source of the RNA samples in the main text - e.g. " RNAs were collected from whole greenhouse grown seedlings" - and more details should be provided in the supplementary material about collecting dates, time of day, numbers of plants and exactly how material was collected and homogenized or pooled.
3. LL 88-98 - The comparison between lupins and cichlids seems awkward to me. Perhaps it would flow better to clearly describe the pattern of diversification in the lupins, pose the hypothesis that adaptations in ancestral North America populations may have promoted diversification in the Mexican highlands and Andes, and then draw parallels to cichlid diversification.
4. LL101-114 - The main text should include a clear description of how genes were classified as evolving under positive selection. According to the Supplementary Notes, the patterns described in the text and Fig 2 are based on a sites model. What was the range in the proportion of sites evolving under positive selection and what was the range of dN/dS ratios for the positively selected sites. Also, why is Fig 2 a line graph? An XY scatter plot with symbols and perhaps regression lines for each clade may be more appropriate.
5. LL123-129 - As acknowledged, given the variation in methodologies, the reported previous results do not provide clear context for the branch-sites test results described in this study.

6. RRBL analyses - I am concerned that all of the browning motion and OU test analyses are assessing may not really be assessing "overall changes in gene expression values along each branch" but rather differences in gene expression relative to differences in branch lengths. Could the findings be an artifact of the shorter branch lengths (and lower variance in branch lengths) in the Andean clade relative to the rest of the tree? In any case, the authors could indicate that they are using a simple Brownian Motion model on L 151. In addition, the supplemental methods could discuss biological replication within clades if not within species.

7. In the end, I am left wondering why genome-wide shifts in molecular evolution and overall gene expression levels should contribute to increased diversification. Are shifts in some genes having manifold effects across the genome? Are positive selection and shifts expression driving adaptations to new environments? If so, are all genes for which shifts have been detected contributing equally to adaptation to new niches? Are GO classifications showing a higher proportion of genes exhibiting signals for adaptive protein evolution or shifts in expression? How much niche-space variation is there among Andean lupin species

Reviewer #2 (Remarks to the Author):

Nevado et al present a fascinating analysis linking macroevolutionary diversification to differential patterns of coding sequence evolution. Using a gene expression data across slow- and fast- diversifying lupins, they show that the fastest diversifying lineage of lupins also show a stronger percentage of loci under positive selection compared to more slowly diversifying lineages. They also find evidence for a shift in the optimal number of gene expression patterns that corresponds to the origin of a putative adaptive radiation.

I think this is an important study with intriguing and provocative results that potentially marks an important step forward in relating genetic change to macroevolutionary diversification. The writing is clear and engaging. My main suggestions concern the diversification analysis. Is it possible to be more precise in assigning the missing species? If so the analyses would be strengthened if the lineage specific option of BAMM were used. Also is it really not possible to look at absolute versus relative ages in dating the tree? You cite other papers that have dated this tree in lines 46-47. Why not use those ages to constrain your analysis here? The implications of this study would be further strengthened if we had a better idea of the timescale of divergence for these clades (and absolute evolutionary rates for gene expression etc).

Authors' response:

Reviewer #1 (Remarks to the Author):

Nevado and coauthors present some very interesting data and patterns of shifts in molecular evolution and gene expression in relation to shifts in diversification. In my opinion, however, the descriptions of the data, analyses and conclusion - "our results represent the first unambiguous evidence for lineage-wide accelerated protein-coding evolution concomitant with the onset of rapid evolutionary radiation" - provided in the main text of the submitted manuscript is too vague for most readers to comprehend and evaluate the implications of the findings. The data and analyses are fairly well described in the Supplementary Notes file, but I think there are some important points should be moved from the Supplementary Notes to the main text. Further, it seems to me that the authors could provide the readers with a better understanding of the implications of their findings.

RE: In the revised version of the manuscript we moved many points from supplementary to the main text, strengthened the conclusions and expanded the discussion to better explain the implications of our findings.

I offer the following more specific comments/suggestions:

1. The authors do not comment on the possible influences of natural hybridization, polyploidy or even gene duplication on their findings. Could any of these processes be contributing to shifts in diversification, molecular evolution or gene expression levels?

RE: These processes can indeed play a role in rapid diversification, and we added this point to the discussion (lines 313-325), but it is notable that there is virtually no variation in ploidy across the western New World *Lupinus* clade, which is the primary focus of the analyses presented here.

2. ~L60 - Readers will want to know what tissues and developmental stages were used for RNA Seq analyses without having to go to the Supplementary Notes. Even after consulting the Supplementary Notes, it is not clear to me that the data are easily comparable across species. I suggest the authors make a simple statement about the source of the RNA samples in the main text - e.g. " RNAs were collected from whole greenhouse grown seedlings" - and more details should be provided in the supplementary material about collecting dates, time of day, numbers of plants and exactly how material was collected and homogenized or pooled.

RE: We added the requested information to both Methods (lines 330-332) and Supplementary Notes (Section 1, second paragraph). Whenever possible we used actively growing young shoots with young leaves and flowerbuds to maximise the number of genes expressed. However, not every species flowered in the greenhouse and for these species we used actively growing shoots with leaves without flowering buds (Supplementary Data 1). To test whether sampling of tissues affected our results we conducted analyses in sub-sets of species with the same type of tissue analysed (only leaves and stem without flowers, or leaves, stem and flower buds), which yielded identical results regardless of tissues used (lines 204-208 and 218-220, and Supplementary Figs. 8 and 9). Thus, we are confident that the results obtained are robust to slight differences in tissue sampling between the species.

3. LL 88-98 - The comparison between lupins and cichlids seems awkward to me. Perhaps it would flow better to clearly describe the pattern of diversification in the lupins, pose the hypothesis that adaptations in ancestral North America populations may have promoted diversification in the Mexican highlands and Andes, and then draw parallels to cichlid diversification.

RE: Following the reviewer' suggestion we describe the patterns of diversification in lupins in much more detail in the revision. We also rewrote the paragraph about cichlids and moved it to discussion (lines 295-312).

4. LL101-114 - The main text should include a clear description of how genes were classified as evolving under positive selection.

RE: We now include a clear description of how genes were classified in the Methods section (lines 356-366).

According to the Supplementary Notes, the patterns described in the text and Fig 2 are based on a sites model. What was the range in the proportion of sites evolving under positive selection and what was the range of dN/dS ratios for the positively selected sites.

RE: We now include the distribution of dN/dS ratios and the proportion of sites under selection in supplementary Fig. 6, and add relevant text to results section (lines 156-162). However, we focus our analyses on the LRT results because the

estimates for the proportion of sites under selection and dN/dS for these sites are relatively imprecise, while the LRT is robust even if accurate parameter estimates are not possible (e.g., Anisimova et al., 2001, Mol Biol Evol 18: 1585-1592).

Also, why is Fig 2 a line graph? An XY scatter plot with symbols and perhaps regression lines for each clade may be more appropriate.

RE: We adapted Fig2B as suggested.

5. LL123-129 - As acknowledged, given the variation in methodologies, the reported previous results do not provide clear context for the branch-sites test results described in this study.

RE: The variation in methodology in the previous studies made it necessary for us to include comparisons with other groups using exactly the same methodology, which supported that adaptation in the Andean and NAP lupins is unmatched in any other plant groups we analysed. We mention the results of the previous studies only to put our results in the context of what's known about molecular adaptation in other organisms.

6. RRBL analyses - I am concerned that all of the browning motion and OU test analyses are assessing may not really be assessing "overall changes in gene expression values along each branch" but rather differences in gene expression relative to differences in branch lengths.

RE: RRBL measures the relative divergence in sequences and expression values. We now clarified this providing more details (lines 191-194 and lines 380-389). The OU test, on the other hand, is a widely used test to detect selection-driven changes to phenotypic traits, including gene expression values (see e.g. Bedford & Hart, 2009, Proc Natl Acad Sci USA 4: 1133-1138; Rohlf et al., 2014, Mol Biol Evol 31: 201-211).

Could the findings be an artifact of the shorter branch lengths (and lower variance in branch lengths) in the Andean clade relative to the rest of the tree?

RE: The clustering of positive RRBL values in the Andean clade is unlikely to be an artefact of the shorter branch lengths, because the change in gene expression in the Andean clade is faster than the simulated values (marked "BM" on figure 3). The simulation used the observed tree topology and branch lengths to obtain the expected change in gene expression values under Brownian motion model (no selection). Thus, the shorter branch lengths in the Andean clade are taken into account in the simulation, yet, the observed data for the Andean clade shows faster than the expected neutral change. We now clarified this in the main text (lines 200-204).

In any case, the authors could indicate that they are using a simple Brownian Motion model on L 151.

RE: Added this information to the Methods section (lines 397-400) and mention it in the results (lines 194-197).

In addition, the supplemental methods could discuss biological replication within clades if not within species.

RE: We added the information about biological replication within clades, and why multiple accessions for most species are not available (lines 372-379).

7. In the end, I am left wondering why genome-wide shifts in molecular evolution and overall gene expression levels should contribute to increased diversification. Are shifts in some genes having manifold effects across the genome? Are positive selection and shifts in expression driving adaptations to new environments? If so, are all genes for which shifts have been detected contributing equally to adaptation to new niches? Are GO classifications showing a higher proportion of genes exhibiting signals for adaptive protein evolution or shifts in expression? How much niche-space variation is there among Andean lupin species?

RE: We rewrote the introduction and discussion to include the issues raised by the reviewer. It seems most likely that the diversification in New World lupins is driven by adaptation to different ecological niches through changes to both coding sequences and gene expression levels affecting different genes and gene functions. The mountainous areas inhabited by both NAP and particularly Andean lupins are remarkably heterogeneous, presenting a large array of different habitats. While we are not aware of any study quantitatively assessing niche

space variation within Andean lupins, in the discussion of the current manuscript we include a much more detailed description of ecological and morphological diversity within this clade (lines 268-277 and 282-289). We also include an analysis of GO terms for genes showing evidence of positive selection on coding sequences, and significant shifts in gene expression (lines 170-172 and 220-225; and Supplementary Figure 9).

Reviewer #2 (Remarks to the Author):

Nevado et al present a fascinating analysis linking macroevolutionary diversification to differential patterns of coding sequence evolution. Using a gene expression data across slow- and fast- diversifying lupins, they show that the fastest diversifying lineage of lupins also show a stronger percentage of loci under positive selection compared to more slowly diversifying lineages. They also find evidence for a shift in the optimal number of gene expression patterns that corresponds to the origin of a putative adaptive radiation.

I think this is an important study with intriguing and provocative results that potentially marks an important step forward in relating genetic change to macroevolutionary diversification. The writing is clear and engaging. My main suggestions concern the diversification analysis. Is it possible to be more precise in assigning the missing species? If so the analyses would be strengthened if the lineage specific option of BAMM were used. Also is it really not possible to look at absolute versus relative ages in dating the tree? You cite other papers that have dated this tree in lines 46-47. Why not use those ages to constrain your analysis here? The implications of this study would be further strengthened if we had a better idea of the timescale of divergence for these clades (and absolute evolutionary rates for gene expression etc).

RE: We now perform the BAMM using clade-specific missing rates, which returned very similar results (Supplementary Figure 4 and Supplementary Note 4.4). Regarding absolute dating of the tree presented in this work, we did not perform this analysis because our dataset is significantly less comprehensive (taxonomically) than used in previous time-calibrated phylogenies of *Lupinus* (e.g. Drummond et al., 2012, *Syst Biol* 61: 443-460) and lacks the necessary outgroup sequences to use fossil information for node calibration. This would force us to use a secondary (effectively tertiary) constraint to fix the age of the New World clade, which together with the relatively sparse taxon sampling within *Lupinus* would result in divergence time and diversification rates estimates that are likely to be far less reliable than those already published (for the effect of sparse taxon sampling see e.g., Linder et al., 2005, *MPE* 35: 569-582; Bokma, 2008, *Evolution* 62: 2441-2445; Brock et al., 2011, *Syst Biol* 60: 410-419). However, in the revision we include more detailed information on the timescale of divergence in

New World lupins with reference to previous work (lines 79-89), which provides a much better understanding of the pace and timing of gene expression and coding sequence divergence analysed in this work.

Reviewers' comments:

Reviewer #1 (Remarks to the Author):

The authors have addressed most of the concerns raised in my initial review, but there is a lingering issue that should be addressed with small, but important changes to the text. The authors imply that rejecting an LRT null hypothesis in favor of differences in selection on a foreground versus background branches equates with identification of genes "under selection" (e.g. lines 171 and 173) and the NAA branch - on which the null hypothesis is often not rejected - is interpreted as exhibiting the "least selection". I don't think the authors mean to imply rampant neutrality on the NAA branch. The authors are seeing shifts in mode of selection - e.g. purifying to positive in some cases - not loss of selection!

Reviewer #2 (Remarks to the Author):

I thought this paper was a strong contribution during my initial review and I think the authors have done a good job of addressing my concerns. I would be happy to see this in Nature Communications and think it marks an important step forward in the study of adaptive radiation.

Authors' response:

Reviewer #1 (Remarks to the Author):

The authors have addressed most of the concerns raised in my initial review, but there is a lingering issue that should be addressed with small, but important changes to the text. The authors imply that rejecting an LRT null hypothesis in favor of differences in selection on a foreground versus background branches equates with identification of genes "under selection" (e.g. lines 171 and 173) and the NAA branch - on which the null hypothesis is often not rejected - is interpreted as exhibiting the "least selection". I don't think the authors mean to imply rampant neutrality on the NAA branch. The authors are seeing shifts in mode of selection - e.g. purifying to positive in some cases - not loss of selection!

RE: We rephrased this sentence (l. 170) to make it clear that we refer to an increase in the frequency of positive selection.

Reviewer #2 (Remarks to the Author):

I thought this paper was a strong contribution during my initial review and I think the authors have done a good job of addressing my concerns. I would be happy to see this in Nature Communications and think it marks an important step forward in the study of adaptive radiation.